# Improving Vision Model Robustness against Misclassification and Uncertainty Attacks via Underconfidence Adversarial Training

Josué Martínez-Martínez[1,2], John Holodnak[1], Olivia Brown[1], Sheida Nabavi[2], Derek Aguiar[2], and Allan Wollaber[1]

[1]MIT Lincoln Laboratory
[2]University of Connecticut
[1]{josue.martinez-martinez, john.holodnak, olivia.brown, allan.wollaber}@ll.mit.edu
[2]{josue.martinez-martinez, sheida.nabavi, derek.aguiar}@uconn.edu

## Abstract

Adversarial robustness research has focused on defending against misclassification attacks. However, such adversarially trained models remain vulnerable to *underconfidence adversarial attacks*, which reduce the model's confidence without changing the predicted class. Decreased confidence can result in unnecessary interventions, delayed diagnoses, and a weakening of trust in automated systems. In this work, we introduce two novel underconfidence attacks: one that induces ambiguity between a class pair, and **ConfSmooth** which spreads uncertainty across all classes. For defense, we propose **Underconfidence Adversarial Training (UAT)** that embeds our underconfidence attacks in an adversarial training framework. We extensively benchmark our underconfidence attacks and defense strategies across six model architectures (both CNN and ViT-based), and seven datasets (MNIST, CIFAR, ImageNet, MSTAR and medical imaging). In 14 of the 15 data-architecture combinations, our attack outperforms the state-of-the-art, often substantially. Our UAT defense maintains the highest robustness against all underconfidence attacks on CIFAR-10, and achieves comparable to or better robustness than adversarial training against misclassification attacks while taking half of the gradient steps. By broadening the scope of adversarial robustness to include uncertainty-aware threats and defenses, UAT enables more robust computer vision systems.

## 1 Introduction

Deep neural networks have achieved state-of-the-art performance in various computer vision tasks, including image classification. Despite these successes, the broad adoption of deep learning models has simultaneously increased their exposure to sophisticated security threats, particularly adversarial attacks [1, 2]. Adversarial attacks introduce imperceptible perturbations in the input data, causing models to produce incorrect predictions. These threats are particularly concerning for safety-critical applications, such as medical diagnosis [3, 4], where errors can have severe consequences. The risk is further amplified in collaborative human-AI settings [5], where incorrect model outputs can compromise trust and effectiveness. As a result, adversarial machine learning has attracted significant attention in the past decade, with over 2,300 studies dedicated to understanding and mitigating these vulnerabilities [6].

However, most adversarial machine learning research focuses on perturbations that induce misclassification. Other vulnerabilities, such as uncertainty manipulation, remain underexplored [7]. These attacks alter confidence scores without changing the predicted class [8], degrading the model calibration and prediction reliability. Poor calibration is especially problematic in high-stakes applications, where over- or under-confidence can lead to unnecessary interventions, redundant oversight, or increased operational costs.

Recent work highlights the potential impact of uncertainty-based attacks, revealing that existing defenses are vulnerable and, counterintuitively, that white-box variants can be less effective than black-box ones due to unintended decision boundary crossings [9]. To address these limitations, we propose *ConfSmooth*, a novel attack that lowers confidence while preserving the original prediction of the model, revealing vulnerabilities missed by accuracy-based metrics. We also introduce *Underconfidence Adversarial Training (UAT)*, a defense designed to counter confidence manipulation while maintaining robustness to misclassification attacks, requiring only half of the gradient steps of standard adversarial training. These methods reveal key vulnerabilities in confidence-based threats and offer practical defenses for high-stakes settings.

## 2 Related work

### 2.1 Adversarial attacks

**Misclassification attacks.** Adversarial robustness research has focused on attacks that induce mis-

Proceedings of the 7th Northern Lights Deep Learning Conference (NLDL), PMLR 307, 2026.

classification. Among early efforts in this area, the fast gradient sign method (FGSM) is a foundational adversarial white-box attack method that perturbs inputs by taking a single step in the direction of the gradient of the loss function, scaled by a small factor [10]. Building on FGSM, projected gradient descent (PGD) performs multiple iterative updates, projecting the perturbed inputs back onto the allowed perturbation set (e.g., a $\ell_\infty$ ball) after each step [11]. The PGD attack is widely considered a strong first-order approach and serves as a standard for evaluating robustness. Other optimization-based attacks have also been developed with the aim of generating minimal and highly effective perturbations to induce misclassifications. The Carlini & Wagner (C&W) attack formulates adversarial example generation as an unconstrained optimization problem, minimizing perturbation magnitude through a tailored loss function [12]. The DeepFool attack approximates the decision boundary of a classifier with a linear function and computes the minimal perturbation required to cross the boundary [13].

**Uncertainty attacks.** Recent work has begun to examine adversarial attacks against prediction confidence. One of the first uncertainty attacks focused on reducing model confidence in the true class while simultaneously increasing confidence in the erroneous classes [8]. The attack adapts PGD by taking steps in the direction of gradient loss using the softmax score associated with the predicted class. Recently, both white-box and black-box attacks based on PGD and Square Attack [14] have been developed with the aim of increasing or decreasing confidence in model predictions [9]. The four white-box variations are: (a) an overconfidence attack that aims to increase the confidence of model predictions; (b) an underconfidence attack, which reduces the confidence of model predictions by reducing the different between the top two predicted class probabilities; (c) a maximum miscalibration attack that reduces the confidence of correct predictions and increases the confidence of incorrect predictions; and (d) a random confidence attack, which randomly applies underconfidence or overconfidence attacks. Guided by its relevance to critical applications, we focus on underconfidence attacks in this work, which can make AI-assisted diagnostics unreliable and complicate decision-making processes.

## 2.2 Adversarial defenses

**Misclassification attack defenses.** Adversarial training (AT) is a foundational defense strategy initially developed to enhance robustness against misclassification attacks by training models on adversarially perturbed examples [10, 11]. Training a model with adversarial examples increases classification accuracy on adversarially perturbed data, but typically reduces clean accuracy. To mitigate this trade-off between clean and robust accuracy, the TRadeoff-inspired Adversarial Defense via Surrogate-loss minimization (TRADES) method combines a classification loss in clean inputs with a Kullback-Leibler (KL) divergence term that penalizes differences between the model predictions in clean and adversarial samples [15]. This loss encourages the model to maintain stable predictions under adversarial perturbations while preserving clean accuracy. RobustAugMix extended this approach by using the Jensen-Shannon (JS) divergence [16], a symmetric and smooth variant of the KL divergence, to measure the similarity between the output distributions of clean, augmented, and adversarial inputs [17, 18]. Although some defenses are designed to jointly improve adversarial robustness and probability calibration, their effectiveness against underconfidence attacks remains unexplored.

**Uncertainty attack defenses.** A recent study evaluated a range of calibration methods as potential defenses against uncertainty attacks [9]. These methods aim to align predicted class probabilities with true class likelihoods and fall into two main categories: post-calibration and training-based calibration. Post-calibration techniques, such as Platt scaling [19], adjust model predictions after training, while training-based approaches incorporate calibration objectives during learning, using bias correction or regularization [20, 21]. Despite their widespread use, neither class of methods demonstrated robustness against uncertainty manipulation attacks [22–25]. An approach designed to improve generalization against adversarial examples is Confidence-Calibrated Adversarial Training (CCAT) [26], which rejects low-confidence adversarial samples during training. Unfortunately, CCAT does not guarantee robustness against underconfidence attacks. To address this limitation, Calibration Attack Adversarial Training (CAAT) was proposed, applying both overconfidence and underconfidence perturbations to each input and minimizing the sum of cross-entropy losses over the resulting examples [9]. However, CAAT-trained models still fail to achieve robustness against uncertainty-based threats.

## 3 Methodology

Our main objective is to identify an adversarial training defense approach robust to underconfidence attacks without significantly compromising robustness against misclassification attacks or clean data performance. To achieve this, we first examine the limitations of calibration attacks and defense strategies [9]. Addressing these weaknesses, we introduce two key modifications to strengthen the attack, which ultimately leads to a more robust defense model.

## 3.1 Underconfidence attack

White-box calibration attacks use PGD to minimize a loss function that encodes the difference between the probabilities assigned to the predicted class (the largest probability) and the next most likely class (the second largest probability) [9]. The loss function is

$$UC(x, \hat{y}) = p_\theta(x)_{\hat{y}} - \max_{j \neq \hat{y}} p_\theta(x)_j, \qquad (1)$$

where $p_\theta(x)_{\hat{y}}$ is the probability of the most confident class and $\max_{j \neq \hat{y}} p_\theta(x)_j$ is the probability of the second most confident class. By minimizing this objective, the attack reduces the probability gap between the two main predictions, making the model appear *underconfident* in its predictions. However, the attack performs better when a dropout mask is applied to the generated perturbations ($\delta$) [9]. Let $m$ be a dropout mask defined as:

$$m_i = \begin{cases} 1, & \text{if } u_i < p, \\ 0, & \text{otherwise}, \end{cases} \qquad (2)$$

where $u_i \sim \text{Uniform}(0, 1)$ and $p$ is the fraction of elements retained. Then, the dropout-modified perturbation is

$$\delta_{\text{dropout}} = \delta \odot m, \qquad (3)$$

where $\odot$ denotes the element-wise product.

This approach may induce oscillations between the probabilities of the top two classes, as it lacks a mechanism to prevent repeated crossings of the decision boundary. Furthermore, it focuses solely on the two most probable classes, disregarding the remaining $K - 2$ classes in multiclass classification settings. To address these limitations, we propose two modifications to the attack: (1) an adaptive step size rule to mitigate oscillations, and (2) a redesigned loss function that encourages a near-uniform distribution over class probabilities, thereby ensuring broader class awareness during optimization.

## 3.2 Adaptive step size with step back

The first limitation that we will address is the constant misclassification that occurs when the underconfidence attack is implemented using PGD. Since this attack approximates a strong adversarial perturbation by repeatedly taking small steps within a constrained norm ball, we propose a modified version in which the step size is adaptively reduced by half whenever the adversarial perturbation crosses the decision boundary, causing a misclassification. Importantly, if a misclassification occurs, we revert the perturbation step before halving the step size, ensuring that the subsequent step restarts from a correctly classified point, which also decreases the confidence. Since halving the step size can be too aggressive, we slightly increase the step size by multiplying it by a factor $\lambda$, which we have empirically

set as 1.1. Formally, this modified PGD update rule is:

$$x^{(t+1)} = \Pi_{x+S}\left(x^{(t)} + \alpha^{(t)} \cdot \text{sign}(\nabla_x \mathcal{L}(\theta, x^{(t)}, \hat{y}))\right), \qquad (4)$$

where $x^{(t)}$ is the adversarial example in step $t$, $\alpha^{(t)}$ is the adaptive step size, $\text{sign}(\cdot)$ is the sign function that returns the element-wise direction of the gradient, $\nabla_x \mathcal{L}(\theta, x^{(t)}, \hat{y})$ is the gradient of the loss $\mathcal{L}(\cdot)$ with respect to the input $x^{(t)}$, calculated for the model parameters $\theta$ and the initial predicted class $\hat{y}$, and $\Pi_{x+S}(\cdot)$ is the projection operator ensuring $x^{(t+1)}$ stays within the allowed perturbation set $S$.

Let $\mathcal{I}^{(t+1)} := (f(x^{(t+1)}) = \hat{y}) \wedge (\mathcal{L}(\theta, x^{(t+1)}, \hat{y}) < \mathcal{L}(\theta, x^{(t)}, \hat{y}))$ denote a consistent classification at step $t + 1$ where the loss function improves. Then, the adaptive step size is defined as:

$$\alpha^{(t+1)} = \begin{cases} \dfrac{\alpha^{(t)}}{2} & \text{if } f(x^{(t+1)}) \neq \hat{y} \\ \alpha^{(t)} \cdot \lambda & \text{if } \mathcal{I}^{(t+1)} \\ \alpha^{(t)} & \text{otherwise} \end{cases} \quad \text{with } \lambda > 1 \qquad (5)$$

and the updated data point is defined as:

$$x^{(t+1)} = \begin{cases} x^{(t)}, & \text{if } f(x^{(t+1)}) \neq \hat{y} \\ x^{(t+1)}, & \text{otherwise}. \end{cases} \qquad (6)$$

This PGD update rule leads the attack to decrease both the number of misclassifications and model confidence. However, this interpolates between two classes (an incorrect and correct class), whereas an ideal uncertainty attack would allocate class confidence uniformly.

## 3.3 ConfSmooth

To overcome the limitation of prior uncertainty attacks that focus on two classes, we propose ConfSmooth, an underconfidence attack that combines cross-entropy loss with label smoothing. This formulation enables us to smooth the classifier-predicted probability distribution across all $K$ classes [27]. We define the loss function as:

$$H_{\text{LS}}(p, q) = -\sum_{i=1}^{K} \left[(1 - \epsilon) \cdot p_i + \frac{\epsilon}{K}\right] \cdot \log(q_i), \qquad (7)$$

where $p_i$ is the $i$-th element of the original one-hot encoded ground truth label distribution (equal to 1 for the initially predicted class and 0 for all other classes), and $q_i$ is the predicted probability of the classifier for class $i$ after adding the attack perturbation. The smoothing parameter $\epsilon \in [0, 1]$ controls the degree of smoothing applied to the ground truth distribution by modifying the ground-truth label distribution from a one-hot encoding (probability 1

**Algorithm 1** Underconfidence attack with adaptive steps

---

**Input:** Classifier $f$, input $x$, initial predicted label $\hat{y}$, step size $\alpha$, perturbation budget $S$, gradient steps $T$, loss function $\mathcal{L}$, increase factor $\lambda > 1$

**Output:** Adversarial example $x^{\text{adv}}$

1: $x^{\text{adv}} \leftarrow x$, $\alpha^{(0)} \leftarrow \alpha$
2: $l_{\text{adv}} \leftarrow \mathcal{L}(f(x^{\text{adv}}), \hat{y})$
3: **for** $t = 0$ to $T - 1$ **do**
4:   Compute gradient $\nabla_x \mathcal{L}(f(x^{adv}), \hat{y})$
   $x^{\text{temp}} \leftarrow \Pi_{x+S}\Big(x^{\text{adv}} + \alpha^{(t)} \operatorname{sign}(\nabla_x \mathcal{L}(f(x^{\text{adv}}), \hat{y}))\Big)$
5:   $l_{\text{temp}} \leftarrow \mathcal{L}(f(x^{\text{temp}}), \hat{y})$
6:   **if** $f(x^{\text{temp}}) \neq \hat{y}$ **then**
7:     $\alpha^{(t+1)} \leftarrow \alpha^{(t)}/2$
8:     $x^{\text{temp}} \leftarrow x^{\text{adv}}$, $l_{\text{temp}} \leftarrow l_{\text{adv}}$
9:   **else if** $f(x^{\text{temp}}) = \hat{y}$ and $l_{\text{temp}} < l_{\text{adv}}$ **then**
10:    $\alpha^{(t+1)} \leftarrow \alpha^{(t)} \lambda$
11:    $x^{\text{adv}} \leftarrow x^{\text{temp}}$, $l_{\text{adv}} \leftarrow l_{\text{temp}}$
12:   **else**
13:    $\alpha^{(t+1)} \leftarrow \alpha^{(t)}$
14:   **end if**
15: **end for**
16: **return** $x^{\text{adv}}$

---

for the initially predicted class and 0 for all others) to a smoother distribution. The resulting probability of the initially predicted class is reduced from 1 to $(1 - \epsilon) + \frac{\epsilon}{K}$, and each incorrect class probability target increases from 0 to exactly $\frac{\epsilon}{K}$.

The ConfSmooth attack algorithm uses PGD (Equation (4)) with an adaptive step size (Equation (5)) and adversarial perturbation update (Equation (6)) to generate adversarial examples (Algorithm 1). By minimizing this objective, ConfSmooth addresses the limitation of Equation (1), which only reduces the gap between the top two predicted classes.

### 3.4 Underconfidence adversarial training (UAT)

Although no defense algorithm has shown high robustness against underconfidence attacks, AT has provided the strongest defense so far [9]. Here, we propose *underconfidence adversarial training (UAT)*, which integrates the robust optimization framework of AT [11] with underconfidence attacks to preserve the predicted label while reducing its confidence. The optimization is given by:

$$\min_{\theta} E_{(x,y) \sim D} \left[ \max_{||\delta||_p < S} L(f_\theta(x + \delta), y) \right], \quad (8)$$

where $\theta$ are the model parameters, $L(\cdot)$ is the loss function (e.g., a cross-entropy loss), the expectation $E_{(x,y) \sim D}$ is taken over the data distribution $D$ of the input-output pairs $(x, y)$, and $\| \cdot \|_p$ denotes an $\ell_p$ norm (typically the $\infty$-norm). The parameter $S$ defines the perturbation budget, restricting the magnitude of adversarial perturbations within the specified norm. The inner maximization approximates

the worst-case perturbation $\delta$ within the constraint $\|\delta\|_p \leq S$ (i.e., maximize the loss) and is typically approximated using PGD (Equation (4)); the outer minimization optimizes the model parameters $\theta$ to minimize the approximation of the worst-case loss.

The main difference between UAT and AT is in the generation of $\delta$. Instead of computing an optimal perturbation to induce a misclassification using a cross-entropy loss like AT, UAT uses Equation (1) or Equation (7), which is an approximation of the optimal perturbation to reduce the confidence of the model. By training the model using the loss functions in Equations (1) or (7) with PGD using an adaptive step size, the model should achieve robustness against underconfidence attacks.

## 4 Experimental Design

We evaluated the effectiveness of our adaptive and ConfSmooth attacks and UAT defense across various model architectures and datasets. To assess the transferability of attacks and model-specific vulnerabilities, we considered convolutional neural networks such as LeNet [28], EfficientNet-B0 [29], ResNet-18 and ResNet-50 [30], WideResNet-50 [31], as well as a vision transformer model, ViT-B16 [32]. We evaluated generalization across domains and task complexity by conducting experiments on seven datasets. Breast Cancer Ultrasound (2 classes) [33], Chest X-ray (3 classes) [34], MNIST (10 classes) [35], CIFAR-10 (10 classes)[36], MSTAR (11 classes) [37, 38], CIFAR-100 (100 classes) [36], and ImageNet (1000 classes) [39]. These datasets span handwritten digits, medical, natural, and radar imagery, providing a diverse testbed for evaluating attack performance and defensive robustness.

### 4.1 Model training

All trained models used stochastic gradient descent (SGD) with Nesterov momentum [40], an initial learning rate of 0.1, and a weight decay of $5 \times 10^{-4}$, following [41]. For CIFAR-10, we trained modified ResNet architectures from scratch for 200 epochs using a cosine learning rate schedule [42], adapting the input convolution kernel size and stride for $32 \times 32$ images [43]. For MNIST, LeNet model was trained from scratch for 5 epochs. For ImageNet, we used the standard pre-trained PyTorch models without additional training. For all other datasets, including breast cancer ultrasound, chest radiography, and radar, we performed transfer learning using ImageNet pretrained backbones, freezing all but the classification head, which is trained for 10 epochs. In addition, images were resized to $224 \times 224$ and augmented with random crops and horizontal flips.

Adversarially trained models followed the same configurations and used multiple PGD variants with 10 gradient steps (5 for UAT-Smooth with a smoothing parameter ($\epsilon$) of 0.98), an initial step size of

| Architecture | Dataset | Conf. (Before) | Conf. - Dropout | Conf. - Adaptive | Conf. - ConfSmooth |
|---|---|---|---|---|---|
| **EfficientNet-B0** | Breast Cancer Ultrasound | $0.61 \pm 0.016$ | $0.52 \pm 0.003$ | $\mathbf{0.50 \pm 0.0001}$ | $0.51 \pm 0.0004$ |
| | Chest X-ray | $0.86 \pm 0.004$ | $\mathbf{0.45 \pm 0.028}$ | $0.61 \pm 0.009$ | $0.51 \pm 0.023$ |
| | MSTAR | $0.51 \pm 0.005$ | $0.39 \pm 0.009$ | $0.31 \pm 0.005$ | $\mathbf{0.15 \pm 0.001}$ |
| | CIFAR-100 | $0.60 \pm 0.002$ | $0.52 \pm 0.004$ | $0.27 \pm 0.002$ | $\mathbf{0.11 \pm 0.001}$ |
| | ImageNet | $0.71$ | $0.12$ | $0.24$ | $\mathbf{0.06}$ |
| **ResNet-50** | Breast Cancer Ultrasound | $0.63 \pm 0.023$ | $0.51 \pm 0.001$ | $\mathbf{0.50 \pm 0.001}$ | $0.51 \pm 0.0003$ |
| | Chest X-ray | $0.78 \pm 0.006$ | $0.48 \pm 0.010$ | $0.45 \pm 0.007$ | $\mathbf{0.38 \pm 0.003}$ |
| | MSTAR | $0.35 \pm 0.004$ | $0.23 \pm 0.002$ | $0.23 \pm 0.004$ | $\mathbf{0.14 \pm 0.007}$ |
| | CIFAR-100 | $0.55 \pm 0.003$ | $0.45 \pm 0.004$ | $0.25 \pm 0.002$ | $\mathbf{0.10 \pm 0.001}$ |
| | ImageNet | $0.40$ | $0.04$ | $0.10$ | $\mathbf{0.03}$ |
| **ViT-B16** | Breast Cancer Ultrasound | $0.82 \pm 0.013$ | $0.52 \pm 0.003$ | $\mathbf{0.50 \pm 0.00001}$ | $0.51 \pm 0.0005$ |
| | Chest X-ray | $0.92 \pm 0.004$ | $0.54 \pm 0.007$ | $0.49 \pm 0.001$ | $\mathbf{0.40 \pm 0.004}$ |
| | MSTAR | $0.60 \pm 0.011$ | $0.37 \pm 0.006$ | $0.34 \pm 0.002$ | $\mathbf{0.23 \pm 0.003}$ |
| | CIFAR-100 | $0.83 \pm 0.002$ | $0.45 \pm 0.003$ | $0.39 \pm 0.001$ | $\mathbf{0.20 \pm 0.002}$ |
| | ImageNet | $0.77$ | $0.30$ | $0.31$ | $\mathbf{0.11}$ |

**Table 1. Mean MSP before/after underconfidence attacks across architectures and datasets.** Lower post-attack values indicates stronger suppression. ConfSmooth is the top performer across most setups. Accuracy and ECE are reported in Table S1.

5/255 and an $L_\infty$ bounded perturbation budget of 8/255. For CAAT, we applied a 95% dropout rate and included only underconfident generated samples, as including overconfident samples does not improve robustness to underconfidence attacks [9]. For RobustAugMix, we included clean and adversarial examples but excluded augmentations that isolate adversarial robustness from generalization. For each architecture, dataset and defense, we trained five independent runs using different random seeds. All training was conducted with a batch size of 128 using two NVIDIA Volta V100 GPUs.

## 4.2 Evaluation setup

We evaluated all models using multiple PGD variants each with 20 gradient steps and an initial step size of 5/255. All perturbations were constrained by the $L_\infty$ norm with a fixed budget of 0.03. The use of more gradient steps ensures better convergence and stronger adversarial examples [44]. We used a recommended dropout factor of 95% for underconfidence attacks leveraging dropout [9]. For ConfSmooth, we set $\epsilon = 0.98$, such that the initially predicted class keeps $(1 - \epsilon) + \frac{\epsilon}{K} = 0.02 + \frac{0.98}{K}$ probability mass, while each other class receives $\epsilon/K$, thereby lowering confidence while preserving the top-1 label. All evaluation numbers are averaged across five independent runs (different random seeds); we report the mean $\pm$ standard deviation.

## 4.3 Metrics

We used five criteria to evaluate model performance and robustness: perturbation magnitude ($\Delta W$), structural similarity index measure (SSIM) [45], classification accuracy, model confidence, and expected calibration error (ECE) [46]. The perturbation magnitude $\Delta W$ is calculated as the Wasserstein distance between a zero vector and the generated perturbation vector [47]. SSIM is calculated between the original image and its adversarial counterpart. Model confidence is defined as the average of the maxi-

mum softmax probabilities (MSP) across evaluated samples.

# 5 Experimental Results

In the adversarial attack results, we refer to the misclassification attack as PGD, the calibration underconfidence attack as Dropout [9], our improved version as Adaptive, and our main contribution as ConfSmooth. In the adversarial defense results, we refer to underconfidence adversarial training using the Adaptive attack as UAT-Gap and using the ConfSmooth attack as UAT-Smooth.

## 5.1 Underconfidence attacks

We evaluated underconfidence attacks across multiple datasets with varying class counts (Table 1). The Dropout attack achieved the highest performance in only a single architecture-dataset configuration, likely due to its fixed step size and dropout factor (Fig. S1). By its use of an adaptive step size, the Adaptive attack largely improved on the performance of the Dropout attack while being more efficient by omitting dropout. However, the Adaptive attack was less effective on large-scale datasets with visually similar categories (e.g., ImageNet-1K). In such cases, underconfidence attacks can leverage the model's tendency to spread the probability mass across many classes [48], identifying regions of high prediction uncertainty.

ConfSmooth outperformed the other attacks except in the breast cancer ultrasound data, where all attacks achieved close to the 50% confidence threshold for a binary classification problem. However, ConfSmooth was not the most performant for EfficientNet trained on chest X-rays, where prediction confidence often remained above 50%. The attack reduced confidence below 50% for most inputs initially predicted as Covid19 or Normal, but largely failed on inputs originally predicted as pneumonia (Fig. S2). Despite incorporating a proposed adaptive step size mechanism, ConfSmooth could not reduce

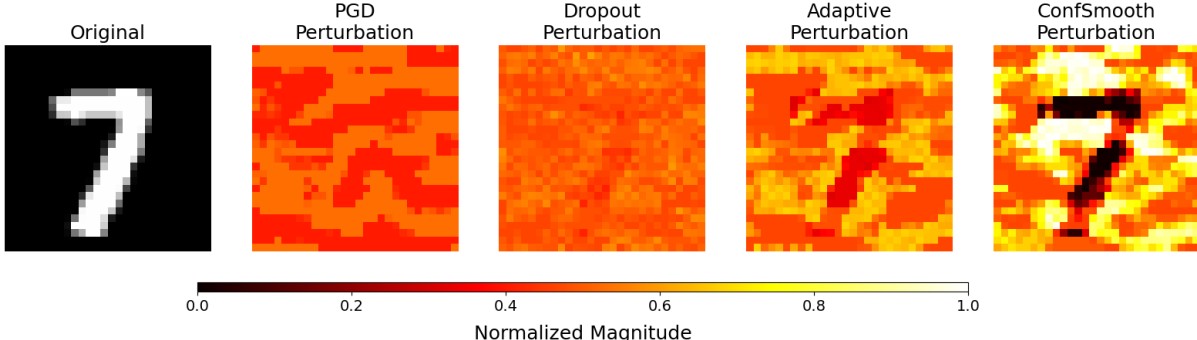

**Figure 1. Original MNIST image alongside perturbation heat-maps for the four PGD attacks variations.** Perturbations (indicated by the horizontal colorbar) are scaled to $[0, 1]$ using the global minimum and maximum perturbation values observed across the four attacks.

confidence without causing misclassification in 603 of $1,288$ samples (47%). This result aligns with previous findings that X-ray images are sensitive to small adversarial changes [49].

Beyond dataset-specific outcomes, we also observed architectural trends. Models based on the ViT architecture were generally more robust to underconfidence attacks than CNN models. This result aligns with previous work on misclassification robustness, which showed that ViTs are more resistant than similarly sized CNNs [50, 51]. ViTs use self-attention to capture global relationships between input patches [52], leading to high-level feature representations that are less sensitive to localized perturbations.

| Attack | $\Delta$W | SSIM |
|---|---|---|
| **PGD** | $0.0274 \pm 0.0034$ | $\mathbf{0.8440 \pm 0.0775}$ |
| **Dropout** | $0.0232 \pm 0.0044$ | $0.7592 \pm 0.1158$ |
| **Adaptive** | $0.0627 \pm 0.0239$ | $0.6813 \pm 0.1197$ |
| **ConfSmooth** | $0.0966 \pm 0.0289$ | $0.6136 \pm 0.1168$ |

**Table 2. Comparison of adversarial attacks on MNIST based on perturbation strength and visual similarity.** $\Delta W$ denotes the average perturbation magnitude (higher is stronger), and SSIM measures structural similarity to the original image (higher is more structural similar). ConfSmooth produces the strongest perturbations, though at the cost of lower SSIM.

## 5.2 Assessing adversarial perturbations

Although underconfidence attacks have been described as harder to detect because they preserve accuracy while requiring smaller perturbations [9], our results reveal a clear trade-off between perturbation strength and structural similarity (Table 2). PGD strikes an effective balance, introducing small perturbations ($\Delta W = 0.0274$) while preserving high structural similarity (SSIM = 0.8440), making it efficient and structure-preserving. The dropout attack produces slightly smaller perturbations ($\Delta W = 0.0232$) but at the cost of a lower SSIM (0.7592). These per-turbations remain small in magnitude and maintain high structural similarity, suggesting that they are harder to detect using standard structural similarity metrics.

In contrast, the Adaptive and ConfSmooth attacks cause more noticeable disruptions to image quality. The Adaptive variant introduces stronger perturbations ($\Delta W = 0.0627$) and visible distortions (SSIM = 0.6813). ConfSmooth is the most aggressive, producing the highest perturbation magnitude ($\Delta W = 0.0966$) and the lowest SSIM (0.6136), leading to a clear degradation in visual fidelity. Interestingly, underconfidence attacks, especially ConfSmooth, tend to concentrate perturbations on pixels less salient to the image class (Figure 1). We attribute this behavior to the constraint of not crossing the decision boundaries, which discourages altering features strongly tied to class prediction. As a result, perturbations often accumulate in background or boundary regions, substantially reducing model confidence while leaving the initial predicted label unchanged.

## 5.3 Robustness to underconfidence attacks

Having previously shown that underconfidence attacks succeed across datasets and architectures (§5.1), we next evaluate whether these attacks remain effective against adversarially trained models. We evaluated adversarial defenses using CIFAR-10, a small-scale dataset with disjoint classes, which provides a clear assessment of robustness while avoiding confounding effects from semantic overlap between classes [48]. Even the least effective attack, Dropout, reduces the confidence of a Vanilla ResNet-18 model to 45% (Table 3). However, the Dropout attack performed poorly against adversarially trained models. The Adaptive attack, which removes dropout and uses an adaptive step size, reduced the confidence of the top class below 50% in most adversarially trained models.

ConfSmooth remained the most effective underconfidence attack overall. On average, ConfSmooth

| Architecture | Defense Method | Conf. (Before) | Conf. - Dropout | Conf. - Adaptive | Conf. - ConfSmooth |
|---|---|---|---|---|---|
| **ResNet-18** | Vanilla | 0.919 ± 0.004 | 0.449 ± 0.006 | 0.414 ± 0.004 | 0.272 ± 0.007 |
| | AT | 0.865 ± 0.002 | 0.674 ± 0.003 | 0.426 ± 0.001 | 0.267 ± 0.002 |
| | TRADES | 0.844 ± 0.044 | 0.668 ± 0.028 | 0.427 ± 0.003 | 0.256 ± 0.003 |
| | RobustAugMix | 0.842 ± 0.003 | 0.654 ± 0.003 | 0.421 ± 0.002 | 0.248 ± 0.001 |
| | CAAT | 0.900 ± 0.006 | 0.645 ± 0.007 | 0.416 ± 0.003 | 0.280 ± 0.011 |
| | UAT-Gap | 0.874 ± 0.016 | 0.688 ± 0.016 | 0.451 ± 0.011 | 0.295 ± 0.013 |
| | UAT-Smooth | 0.935 ± 0.005 | **0.738 ± 0.005** | **0.483 ± 0.003** | **0.367 ± 0.005** |
| **ResNet-50** | Vanilla | 0.945 ± 0.005 | 0.502 ± 0.011 | 0.450 ± 0.008 | 0.345 ± 0.015 |
| | AT | 0.901 ± 0.007 | 0.708 ± 0.007 | 0.442 ± 0.004 | 0.317 ± 0.009 |
| | TRADES | 0.876 ± 0.041 | 0.684 ± 0.032 | 0.452 ± 0.014 | 0.302 ± 0.033 |
| | RobustAugMix | 0.884 ± 0.011 | 0.691 ± 0.008 | 0.436 ± 0.007 | 0.294 ± 0.016 |
| | CAAT | 0.948 ± 0.007 | 0.709 ± 0.012 | 0.454 ± 0.008 | 0.369 ± 0.019 |
| | UAT-Gap | 0.933 ± 0.004 | 0.726 ± 0.003 | 0.496 ± 0.014 | 0.440 ± 0.027 |
| | UAT-Smooth | 0.965 ± 0.004 | **0.767 ± 0.009** | **0.546 ± 0.039** | **0.532 ± 0.054** |
| **WideResNet-50** | Vanilla | 0.947 ± 0.003 | 0.502 ± 0.011 | 0.449 ± 0.005 | 0.344 ± 0.019 |
| | AT | 0.889 ± 0.007 | 0.699 ± 0.008 | 0.429 ± 0.004 | 0.306 ± 0.011 |
| | TRADES | 0.876 ± 0.042 | 0.662 ± 0.039 | 0.450 ± 0.015 | 0.298 ± 0.039 |
| | RobustAugMix | 0.894 ± 0.006 | 0.700 ± 0.004 | 0.439 ± 0.002 | 0.308 ± 0.007 |
| | CAAT | 0.955 ± 0.006 | 0.728 ± 0.012 | 0.453 ± 0.008 | 0.385 ± 0.013 |
| | UAT-Gap | 0.927 ± 0.004 | 0.721 ± 0.005 | 0.514 ± 0.010 | 0.469 ± 0.021 |
| | UAT-Smooth | 0.965 ± 0.004 | **0.768 ± 0.010** | **0.577 ± 0.026** | **0.574 ± 0.035** |

**Table 3. Mean MSP confidence before and after underconfidence attacks on CIFAR-10 across CNN architectures and defenses.** UAT-Smooth maintains the highest post-attack confidence, indicating strong robustness. ECE results are reported in Table S2.

reduced the confidence of adversarially trained models by an additional 10% compared to the Adaptive attack. An exception was our proposed defense, UAT, which substantially improved robustness against underconfidence attacks. UAT-Smooth consistently achieved the highest post-attack confidence across all architectures. For example, on WideResNet-50, UAT-Smooth retained 0.574 confidence after ConfSmooth, compared to 0.344 for the vanilla model and 0.306 for AT. Against the less aggressive dropout-based attack, UAT-Smooth achieved up to 0.768 confidence, outperforming all other defenses. Notably, UAT-Smooth was trained using only half as many attack gradient steps as the other adversarial training methods, demonstrating strong efficiency.

Finally, we observed that many defenses developed for misclassification attacks, such as AT, TRADES, and RobustAugMix, remain vulnerable to underconfidence attacks. This suggests that robustness to misclassification attacks does not imply robustness to underconfidence attacks, and highlights the need for training objectives that explicitly consider model prediction confidence. Adversarially trained models using misclassification objectives tend to be less confident than vanilla models [53]; however, this behavior did not generalize to models trained with underconfidence-based objectives.

To better understand the prediction confidence in response to underconfidence attacks, we examined the confidence on correctly and incorrectly classified samples across different defenses before and after applying the ConfSmooth attack. The vanilla model displayed well-calibrated behavior: low confidence for incorrect predictions and high confidence for correct predictions (Figs. 2(a) and 2(b)). Interestingly, the model trained with CAAT assigned slightly

higher confidence to incorrect predictions than the vanilla model, but still maintained a clear separation between correct and incorrect samples (Figs. 2(c) and 2(d)). In contrast, the model trained with UAT-Smooth assigned noticeably higher confidence to incorrect predictions compared to other methods (Figs. 2(e) and 2(f)). However, it maintained even greater confidence in the correctly classified samples, exhibiting the strongest distinction between the two groups. In summary, the confidence distributions for the vanilla model and model trained with CAAT collapse below the 50% threshold, leading to overlapping distributions, while UAT-Smooth retained a significant portion of its confidence mass for correctly classified samples above 50% with many predictions remaining near 100%.

## 5.4 Robustness to misclassification attacks

Having established robustness to underconfidence attacks, we next assessed how underconfidence attack defenses perform under misclassification attacks (Table 4). We observed that CAAT, trained using only 5% of the perturbation magnitude, offers some robustness against the PGD attack compared to the vanilla model. However, our proposed methods, based on UAT, achieved significantly higher robustness than CAAT, with improvements of at least 12% across the three architectures. Interestingly, UAT-Smooth achieved robustness to misclassiciation attacks comparable to, or exceeding, AT.

The robustness of UAT-Smooth against misclassification attacks emerges from its training objective. Although UAT-Smooth does not directly optimize for misclassification robustness, it uses a PGD-style algorithm to target regions of high model uncer-

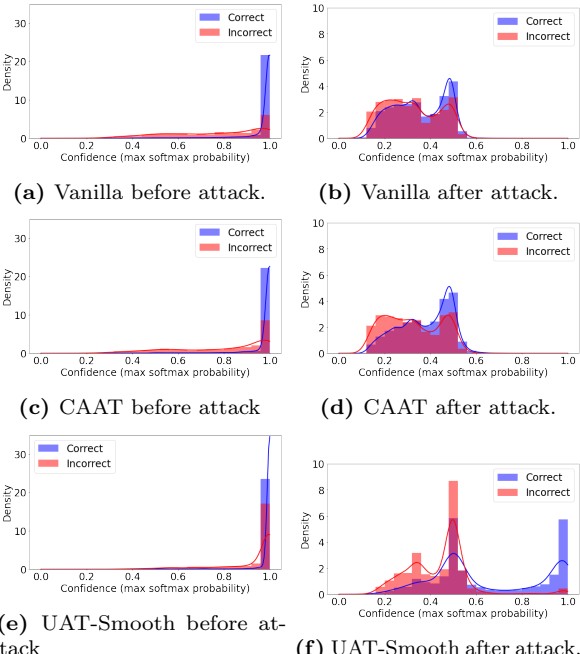

**(a)** Vanilla before attack.

**(b)** Vanilla after attack.

**(c)** CAAT before attack

**(d)** CAAT after attack.

**(e)** UAT-Smooth before attack

**(f)** UAT-Smooth after attack.

**Figure 2. Confidence (MSP) distributions for correct and incorrect samples before and after ConfSmooth perturbation.** Rows correspond to different defense methods (on WideResNet-50) and columns indicated before (left) and after (right) underconfidence attacks. Vanilla and CAAT shows marked confidence drops for correct predictions after being attacked, while UAT-Smooth retains high confidence, indicating stronger robustness to confidence suppression.

| Defense | Accuracy (Before) | Accuracy (After) |
|---|---|---|
| **ResNet-18** | | |
| Vanilla | **0.885 ± 0.004** | 0.000 ± 0.000 |
| AT | 0.758 ± 0.005 | 0.358 ± 0.002 |
| TRADES | 0.733 ± 0.006 | **0.381 ± 0.003** |
| RobustAugMix | 0.742 ± 0.001 | 0.346 ± 0.005 |
| CAAT | 0.852 ± 0.003 | 0.200 ± 0.006 |
| UAT-Gap | 0.657 ± 0.042 | 0.328 ± 0.015 |
| UAT-Smooth | 0.679 ± 0.012 | 0.344 ± 0.007 |
| **ResNet-50** | | |
| Vanilla | **0.882 ± 0.006** | 0.000 ± 0.000 |
| AT | 0.775 ± 0.002 | 0.372 ± 0.005 |
| TRADES | 0.778 ± 0.041 | **0.486 ± 0.137** |
| RobustAugMix | 0.754 ± 0.007 | 0.360 ± 0.004 |
| CAAT | 0.856 ± 0.007 | 0.205 ± 0.008 |
| UAT-Gap | 0.738 ± 0.010 | 0.346 ± 0.006 |
| UAT-Smooth | 0.729 ± 0.009 | 0.384 ± 0.025 |
| **WideResNet-50** | | |
| Vanilla | **0.886 ± 0.004** | 0.000 ± 0.000 |
| AT | 0.768 ± 0.005 | 0.381 ± 0.007 |
| TRADES | 0.773 ± 0.028 | **0.491 ± 0.144** |
| RobustAugMix | 0.753 ± 0.003 | 0.369 ± 0.004 |
| CAAT | 0.865 ± 0.005 | 0.226 ± 0.007 |
| UAT-Gap | 0.721 ± 0.003 | 0.347 ± 0.007 |
| UAT-Smooth | 0.724 ± 0.008 | 0.407 ± 0.029 |

**Table 4. Classification accuracy of CNNs on CIFAR-10 before and after PGD attacks across adversarial defenses.** UAT performs comparably to AT, achieving similar post-attack accuracy on ResNet-50 and WideResNet-50. TRADES achieves the highest overall post-misclassification attack accuracy.

# 6 Conclusion

In this work, we presented two new underconfidence attacks (Adaptive and ConfSmooth) and UAT, an underconfidence adversarial training algorithm that leverages our new attacks. Among the underconfidence attacks we evaluated, ConfSmooth generalized more effectively across datasets and domains, a performance that we attribute to its label-smoothing loss and its adaptive step size to prevent constant misclassifications. Architectural differences also influenced robustness: ViT consistently outperformed CNNs, probably due to their emphasis on global semantics rather than local textures. ConfSmooth reliably reduced the model confidence below the critical threshold of 0.5 (MSP), except when UAT-Smooth was applied. Adversarily trained models optimized for misclassification robustness are paradoxically more susceptible to ConfSmooth than the vanilla model, a vulnerability most pronounced in TRADES, despite its state-of-the-art performance against PGD. Interestingly, UAT-Smooth provided robustness to both underconfidence and misclassification attacks, often matching or exceeding AT while requiring half of the steps. However, we did observe a reduced clean accuracy and increased confidence in incorrect classifications compared to the vanilla model and CAAT. These findings highlight the need for defenses that account not only for prediction correctness but also for calibrated uncertainty under adversarial conditions.

tainty. This strategy exposes weaknesses that help models to generalize better under adversarial conditions. ConfSmooth generates perturbations nearly four times larger than PGD (see §5.2), making the perturbed images more visually distorted. These visual distortions introduce a trade-off, reducing model performance on clean images in our evaluations. UAT-Smooth showed lower clean accuracy than most adversarial defenses (resulting in an additional 8% drop relative to AT on ResNet-18).

In contrast, larger architectures such as ResNet-50 and WideResNet-50 maintained both higher clean accuracy and robustness to misclassification attacks using PGD compared to ResNet-18. For example, on WideResNet-50, UAT-Smooth reached 0.407 post-attack accuracy, outperforming AT (0.381), while maintaining a clean accuracy of 0.724. This suggests that UAT-based defenses scale well with model capacity, allowing a better balance between robustness and accuracy as the size of the architecture increases. However, the extent to which these improvements stem from model capacity rather than properties of UAT itself remains an open question. Additional research is needed to disentangle these effects and identify strategies that preserve clean accuracy without sacrificing robustness.

# 7    Acknowledgment

The authors thank Harry Li, Pooya Khorrami and Sheila Alemany Blanco for their invaluable feedback, which significantly improved this work.

DISTRIBUTION STATEMENT A. Approved for public release. Distribution is unlimited. This material is based upon work supported by the Under Secretary of Defense for Research and Engineering under Air Force Contract No. FA8702-15-D-0001 or FA8702-25-D-B002. Any opinions, findings, conclusions or recommendations expressed in this material are those of the author(s) and do not necessarily reflect the views of the Under Secretary of Defense for Research and Engineering. © 2025 Massachusetts Institute of Technology. Delivered to the U.S. Government with Unlimited Rights, as defined in DFARS Part 252.227-7013 or 7014 (Feb 2014). Notwithstanding any copyright notice, U.S. Government rights in this work are defined by DFARS 252.227-7013 or DFARS 252.227-7014 as detailed above. Use of this work other than as specifically authorized by the U.S. Government may violate any copyrights that exist in this work.

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

# 8 Supplementary Material

## S1 Assessing Dropout attack

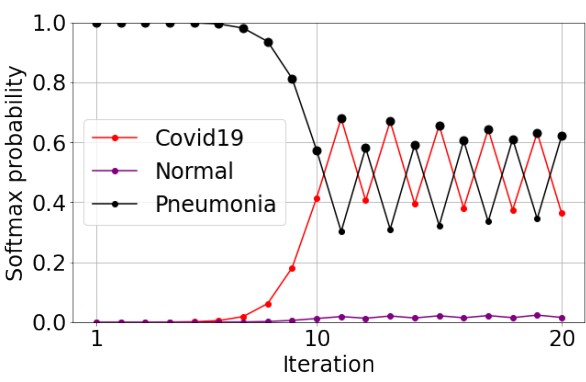

**Figure S1. Interpreting underconfidence attacks that use dropout.** Using dropout, the underconfidence attack reduces the confidence of the Pneumonia class until the decision boundary of the Covid19 class is crossed. At this point, confidence is not further reduced due to oscillations between the two classes.

In our experiments, we found that applying the Dropout attack led to oscillations between the probabilities associated with two classes (Fig. S1). For example, we applied the attack to a three-class problem (chest X-ray dataset) and found that the probability of Pneumonia diminishes until reaching a certain threshold (58%), at which point the attack crosses the boundary and Covid19 becomes the predicted class. In the remaining steps, the attack causes the probabilities of these two classes to oscillate back and forth. It appears that as the attacked point approaches the decision boundary of the model, the perturbations applied may be too large and that limiting them with dropout is not sufficient to avoid large changes in the predicted probabilities.

## S2 Assessing ConfSmooth attack steps during underconfidence adversarial training

We vary the number of ConfSmooth gradient steps taken during UAT-Smooth adversarial training and measure its effect on clean accuracy and robustness to the ConfSmooth under-confidence attack (Table S3). As the steps count increases from 1 to 10, clean accuracy steadily falls from 0.84 to 0.64, while the confidence in pre-attack max-softmax remains high (0.94-0.97). Robustness peaks at five gradient steps: post-attack confidence climbs from 0.38 (1 step) to 0.57 (5 steps) and then declines slightly at higher counts. This trade-off shows that although more steps initially achieve resistance to ConfSmooth, beyond five they erode both clean performance and robustness. In practice, 3–5 steps strike the best balance, matching the standard AT

accuracy and substantially mitigating the ConfSmooth attack.

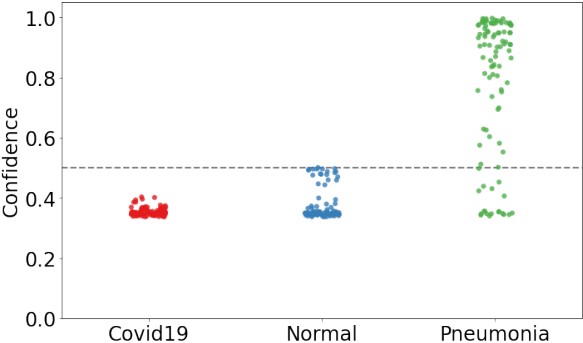

**Figure S2. EfficientNet chest X-ray confidence distributions after ConfSmooth attack** Each point shows the MSP for samples originally predicted as Covid-19 (red), Normal (blue), or Pneumonia (green). The dashed line at 0.5 marks the classifier threshold we want to break with the underconfidence attack. ConfSmooth drives most Covid-19 and Normal confidences below 0.5 but largely fails to suppress Pneumonia predictions.

| Architecture | Dataset | Accuracy | ECE (Before) | ECE - Dropout | ECE - Adaptive | ECE - ConfSmooth |
|---|---|---|---|---|---|---|
| **EfficientNet** | Breast Cancer Ultrasound | $0.75 \pm 0.07$ | $0.16 \pm 0.05$ | $0.23 \pm 0.07$ | $\mathbf{0.25 \pm 0.08}$ | $0.24 \pm 0.07$ |
| | Chest X-ray | $0.90 \pm 0.01$ | $0.05 \pm 0.00$ | $\mathbf{0.46 \pm 0.04}$ | $0.34 \pm 0.01$ | $0.43 \pm 0.02$ |
| | MSTAR | $0.67 \pm 0.01$ | $0.17 \pm 0.01$ | $0.29 \pm 0.01$ | $0.31 \pm 0.01$ | $\mathbf{0.52 \pm 0.01}$ |
| | CIFAR-100 | $0.60 \pm 0.002$ | $0.02 \pm 0.002$ | $0.09 \pm 0.01$ | $0.33 \pm 0.002$ | $\mathbf{0.49 \pm 0.002}$ |
| | ImageNet | $0.79$ | $0.08$ | $0.67$ | $0.55$ | $\mathbf{0.73}$ |
| **ResNet-50** | Breast Cancer Ultrasound | $0.86 \pm 0.00$ | $0.24 \pm 0.02$ | $0.36 \pm 0.002$ | $\mathbf{0.36 \pm 0.001}$ | $0.35 \pm 0.00$ |
| | Chest X-ray | $0.86 \pm 0.02$ | $0.09 \pm 0.01$ | $0.39 \pm 0.03$ | $0.41 \pm 0.03$ | $\mathbf{0.48 \pm 0.02}$ |
| | MSTAR | $0.57 \pm 0.01$ | $0.23 \pm 0.01$ | $0.23 \pm 0.01$ | $0.34 \pm 0.01$ | $\mathbf{0.43 \pm 0.02}$ |
| | CIFAR-100 | $0.58 \pm 0.01$ | $0.04 \pm 0.003$ | $0.14 \pm 0.01$ | $0.33 \pm 0.01$ | $\mathbf{0.48 \pm 0.01}$ |
| | ImageNet | $0.83$ | $0.43$ | $0.79$ | $0.73$ | $\mathbf{0.81}$ |
| **ViT** | Breast Cancer Ultrasound | $0.89 \pm 0.01$ | $0.08 \pm 0.02$ | $0.36 \pm 0.01$ | $\mathbf{0.39 \pm 0.02}$ | $0.38 \pm 0.01$ |
| | Chest X-ray | $0.93 \pm 0.00$ | $0.03 \pm 0.00$ | $0.39 \pm 0.01$ | $0.44 \pm 0.004$ | $\mathbf{0.54 \pm 0.00}$ |
| | MSTAR | $0.78 \pm 0.01$ | $0.18 \pm 0.02$ | $0.41 \pm 0.01$ | $0.44 \pm 0.01$ | $\mathbf{0.55 \pm 0.01}$ |
| | CIFAR-100 | $0.79 \pm 0.003$ | $0.04 \pm 0.002$ | $0.36 \pm 0.003$ | $0.41 \pm 0.003$ | $\mathbf{0.59 \pm 0.003}$ |
| | ImageNet | $0.82$ | $0.06$ | $0.53$ | $0.51$ | $\mathbf{0.71}$ |

**Table S1. ECE before and after underconfidence attacks across datasets and architectures.** Accuracy is presented only before attacks since the underconfidence threats evaluated in this study does not cause misclassification. Higher post-attack ECE indicates stronger miscalibration; bold values denote the maximum post-attack ECE per dataset–architecture setup.

| Architecture | Defense Method | ECE (Before) | ECE - Dropout | ECE - Adaptive | ECE - ConfSmooth | ECE - PGD |
|---|---|---|---|---|---|---|
| **ResNet-18** | Vanilla | $0.037 \pm 0.003$ | $0.436 \pm 0.006$ | $0.470 \pm 0.005$ | $0.613 \pm 0.007$ | $1.000 \pm 0.000$ |
| | AT | $0.109 \pm 0.002$ | $0.090 \pm 0.003$ | $0.333 \pm 0.004$ | $0.491 \pm 0.005$ | $0.543 \pm 0.003$ |
| | TRADES | $0.080 \pm 0.006$ | $0.092 \pm 0.021$ | $0.307 \pm 0.004$ | $0.477 \pm 0.004$ | $\mathbf{0.442 \pm 0.016}$ |
| | RobustAugMix | $0.101 \pm 0.004$ | $0.093 \pm 0.003$ | $0.321 \pm 0.002$ | $0.494 \pm 0.001$ | $0.537 \pm 0.006$ |
| | CAAT | $0.050 \pm 0.004$ | $0.206 \pm 0.005$ | $0.436 \pm 0.002$ | $0.571 \pm 0.009$ | $0.737 \pm 0.009$ |
| | UAT-Gap | $0.217 \pm 0.058$ | $\mathbf{0.068 \pm 0.026}$ | $0.207 \pm 0.053$ | $0.363 \pm 0.054$ | $0.584 \pm 0.021$ |
| | UAT-Smooth | $0.257 \pm 0.012$ | $0.136 \pm 0.017$ | $\mathbf{0.089 \pm 0.019}$ | $\mathbf{0.314 \pm 0.014}$ | $0.669 \pm 0.004$ |
| **ResNet-50** | Vanilla | $0.065 \pm 0.003$ | $0.380 \pm 0.010$ | $0.432 \pm 0.008$ | $0.537 \pm 0.014$ | $1.000 \pm 0.000$ |
| | AT | $0.128 \pm 0.005$ | $0.084 \pm 0.003$ | $0.332 \pm 0.003$ | $0.458 \pm 0.008$ | $0.570 \pm 0.005$ |
| | TRADES | $0.100 \pm 0.017$ | $0.098 \pm 0.029$ | $0.326 \pm 0.029$ | $0.475 \pm 0.016$ | $\mathbf{0.397 \pm 0.132}$ |
| | RobustAugMix | $0.132 \pm 0.008$ | $0.078 \pm 0.006$ | $0.318 \pm 0.007$ | $0.460 \pm 0.013$ | $0.568 \pm 0.015$ |
| | CAAT | $0.094 \pm 0.008$ | $0.149 \pm 0.009$ | $0.402 \pm 0.011$ | $0.487 \pm 0.019$ | $0.772 \pm 0.008$ |
| | UAT-Gap | $0.196 \pm 0.009$ | $\mathbf{0.062 \pm 0.007}$ | $0.242 \pm 0.023$ | $0.298 \pm 0.037$ | $0.623 \pm 0.004$ |
| | UAT-Smooth | $0.237 \pm 0.008$ | $0.103 \pm 0.036$ | $\mathbf{0.129 \pm 0.036}$ | $\mathbf{0.197 \pm 0.049}$ | $0.646 \pm 0.014$ |
| **WideResNet-50** | Vanilla | $0.062 \pm 0.002$ | $0.384 \pm 0.009$ | $0.437 \pm 0.004$ | $0.541 \pm 0.016$ | $1.000 \pm 0.000$ |
| | AT | $0.123 \pm 0.005$ | $0.083 \pm 0.005$ | $0.339 \pm 0.005$ | $0.462 \pm 0.009$ | $0.552 \pm 0.005$ |
| | TRADES | $0.104 \pm 0.021$ | $0.117 \pm 0.052$ | $0.323 \pm 0.018$ | $0.476 \pm 0.013$ | $\mathbf{0.394 \pm 0.134}$ |
| | RobustAugMix | $0.142 \pm 0.006$ | $0.076 \pm 0.003$ | $0.314 \pm 0.005$ | $0.445 \pm 0.006$ | $0.574 \pm 0.007$ |
| | CAAT | $0.091 \pm 0.003$ | $0.139 \pm 0.007$ | $0.412 \pm 0.005$ | $0.479 \pm 0.008$ | $0.755 \pm 0.008$ |
| | UAT-Gap | $0.207 \pm 0.005$ | $\mathbf{0.061 \pm 0.003}$ | $0.207 \pm 0.011$ | $0.252 \pm 0.022$ | $0.625 \pm 0.006$ |
| | UAT-Smooth | $0.241 \pm 0.007$ | $0.101 \pm 0.026$ | $\mathbf{0.130 \pm 0.027}$ | $\mathbf{0.152 \pm 0.030}$ | $0.654 \pm 0.011$ |

**Table S2. ECE before and after underconfidence and misclassification attacks on CIFAR-10 across CNN architectures and defense methods.** For each attack–defense pair, bold values denote the lowest post-attack ECE.

| Steps | Accuracy | Confidence(Before) | Confidence (After) |
|---|---|---|---|
| **One** | $0.84 \pm 0.008$ | $0.96 \pm 0.015$ | $0.38 \pm 0.03$ |
| **Three** | $0.78 \pm 0.004$ | $0.97 \pm 0.004$ | $0.45 \pm 0.04$ |
| **Five** | $0.72 \pm 0.008$ | $0.96 \pm 0.004$ | $0.57 \pm 0.035$ |
| **Seven** | $0.69 \pm 0.012$ | $0.96 \pm 0.006$ | $0.56 \pm 0.040$ |
| **Ten** | $0.64 \pm 0.021$ | $0.94 \pm 0.005$ | $0.49 \pm 0.030$ |

**Table S3. Effect of ConfSmooth gradient steps taken during UAT-Smooth adversarial training (on WideResnet-50).** Taking more than five gradient steps sharply reduces clean accuracy and robustness to ConfSmooth, indicating a trade-off between robustness and standard performance. Accuracy is presented only before attack since ConfSmooth does not cause misclassification.

