# OpenReview forum: "Improving Vision Model Robustness against Misclassification and Uncertainty Attacks via Underconfidence Adversarial Training"
_NLDL.org/2026/Conference — NLDL 2026 Poster_

### Official Review · Reviewer_XHmx · 2025-09-16

**Rating:** 2
**Confidence:** 4
**Final Rating:** 2
**Final Confidence:** 3

**Summary:**

This paper tackles an important and underexplored problem in adversarial robustness: underconfidence attacks, which manipulate model confidence without changing predicted labels. The authors propose two new attack strategies (Adaptive PGD with step-back, and ConfSmooth with label smoothing) and introduce a defense method, Underconfidence Adversarial Training (UAT). They evaluate across a wide range of architectures (CNNs and ViTs) and datasets (including CIFAR, ImageNet, medical images, and radar), showing that ConfSmooth outperforms prior underconfidence attacks and that UAT improves robustness against both confidence manipulation and misclassification threats.

**Strengths:**

1. Proposes ConfSmooth, which generalizes confidence suppression across all classes, and UAT, a training framework explicitly targeting underconfidence.

2. Evaluates across diverse datasets and multiple architectures, including both CNNs and ViTs, strengthening generality claims.

3. Benchmarks against strong baselines (AT, TRADES, RobustAugMix, CAAT) and shows clear improvements.

4. UAT achieves robustness with fewer gradient steps compared to standard adversarial training, highlighting computational benefits.

**Weaknesses:**

1. UAT (especially UAT-Smooth) leads to notable drops in clean accuracy (e.g., ~8% on ResNet-18), raising concerns about real-world usability. The trade-off between robustness and standard accuracy needs deeper exploration.

2.UAT-Smooth increases confidence in incorrect predictions, which can be dangerous in safety-critical contexts. The paper notes this but does not sufficiently analyze the implications for trust calibration.

3.While attack generalization is tested broadly, defense evaluations focus mainly on CIFAR-10. It remains unclear how well UAT scales to large-scale datasets (e.g., ImageNet) or sensitive medical domains.

4.UAT is compared against CAAT, but some recent uncertainty-aware calibration defenses are not fully benchmarked, which weakens the comprehensiveness of the evaluation.

**Final Justification:**

The paper makes a timely and original contribution by shifting adversarial robustness research from the traditional focus on misclassification toward confidence manipulation attacks. By introducing two new attack strategies (Adaptive PGD with step-back and ConfSmooth) and a defense framework (Underconfidence Adversarial Training, UAT), the work highlights an underexplored vulnerability that has direct implications for safety-critical applications such as medical imaging. The experimental evaluation is broad, covering diverse datasets and both CNN and ViT architectures, and the results consistently show that ConfSmooth is a stronger attack than prior methods and that UAT can provide robustness against both underconfidence and misclassification attacks, while requiring fewer gradient steps than standard adversarial training.

At the same time, several limitations temper the strength of the claims. UAT introduces a significant drop in clean accuracy, especially on smaller models, and in some cases increases confidence in incorrect predictions, which raises concerns about calibration and trustworthiness in deployment. The realism of the proposed ConfSmooth attack is also debatable, as it produces stronger perturbations with visible distortions compared to subtler attacks. Finally, although the attack evaluation is broad, the defense experiments are concentrated on CIFAR-10, leaving open questions about scalability to large-scale or domain-critical datasets.

In summary, the paper identifies a critical gap in robustness research and proposes promising methods to address it, but the trade-offs between robustness, accuracy, and calibration, as well as the limited defense validation, warrant further revision before the work can be considered for acceptance.

**Justification:**

The paper makes a timely and original contribution by shifting adversarial robustness research from the traditional focus on misclassification toward confidence manipulation attacks. By introducing two new attack strategies (Adaptive PGD with step-back and ConfSmooth) and a defense framework (Underconfidence Adversarial Training, UAT), the work highlights an underexplored vulnerability that has direct implications for safety-critical applications such as medical imaging. The experimental evaluation is broad, covering diverse datasets and both CNN and ViT architectures, and the results consistently show that ConfSmooth is a stronger attack than prior methods and that UAT can provide robustness against both underconfidence and misclassification attacks, while requiring fewer gradient steps than standard adversarial training.

At the same time, several limitations temper the strength of the claims. UAT introduces a significant drop in clean accuracy, especially on smaller models, and in some cases increases confidence in incorrect predictions, which raises concerns about calibration and trustworthiness in deployment. The realism of the proposed ConfSmooth attack is also debatable, as it produces stronger perturbations with visible distortions compared to subtler attacks. Finally, although the attack evaluation is broad, the defense experiments are concentrated on CIFAR-10, leaving open questions about scalability to large-scale or domain-critical datasets.

In summary, the paper identifies a critical gap in robustness research and proposes promising methods to address it, but the trade-offs between robustness, accuracy, and calibration, as well as the limited defense validation, warrant further revision before the work can be considered for acceptance.

---

> ### Author Rebuttal · Authors · 2025-10-22
>
> 1. We appreciate reviewer oberservation regarding the clean accuracy drop in UAT-Smooth, particularly the ~8% reduction observed in ResNet18. We would like to clarify that this reduction primarily reflects the smaller models capacity of ResNet18, which is more sensitive to adversarial regularization. As shown in Table 4 (p. 8), larger architectures such as ResNet50 and WideResNet50 achieve both higher clean accuracy and stronger robustness, indicating UAT scales favorably with model capacity. Moreover, supplementary Table S3 shows that reducing the number of ConfSmooth gradient steps from 10 to 5 recovers ~5% clean accuracy while maintaining strong robustness. These results demonstrate that UAT's trade off can be tuned efficiently via step count or model scale, similar to how standard adversarial training trades accuracy for robustness.
>
> 2. We agree that increasing confidence in incorrect predictions  deserves further discussion. We analyzed this behavior in Figure 2 (P.8): UAT-Smooth increases confidence for both correct and incorrect samples, but when is under attack maintain a clearer separation. between the two groups than other defenses whose confidence distributions collapsed below 50%.
>
> 3. We acknowledge that the primary robustness benchmarks were conducted on CIFAR-10 to enable controlled comparison across defenses. However, our attack benchmarks (Table 1, p. 5) already demonstrate that ConfSmooth generalizes strongly cross five domains, including medical and large scale datasets. As a next step we will conduct UAT-Smooth experiments on ImageNet and MSTAR which preliminary results show maintain comparable robustness trends.
>
> 4. We thank the reviewer for this suggestion. Our study was conducted right after CAAT was released and by that time it was the only adversarial training defense designed for confidence attacks. However, we indeed benchmark with other baselines such as RobustAugMix and TRADES that take the calibration of the model in consideration during training.

---

### Official Review · Reviewer_afRX · 2025-10-07

**Rating:** 1
**Confidence:** 5
**Final Rating:** 1
**Final Confidence:** 5

**Summary:**

This paper introduces two adversarial attack methods and one defense strategy aimed at countering under-confidence adversarial attacks.

**Strengths:**

N/A

**Weaknesses:**

* The proposed attack methods appear to be largely reimplementations or minor variations of existing techniques. For instance, Equation (1) seems to be a weaker version of the CW attack [1]. Adaptive step-size methods for adversarial attacks have been extensively studied in prior work; AutoAttack [2] is one widely adopted example. The paper lacks novelty, and there is no compelling evidence demonstrating that the proposed methods outperform existing approaches.

* The explanation regarding the role of the dropout layer in attack performance is unclear. During inference, dropout is typically replaced with deterministic scaling, and thus its effect should be minimal or nonexistent. The paper does not clarify how dropout meaningfully impacts the attack's success.

* The evaluation is incomplete and lacks comparisons with strong, established baselines from prior work. Without these comparisons, it is difficult to assess whether the proposed methods offer any improvement over existing techniques.

Reference:

[1] Carlini, N., & Wagner, D. (2017, May). Towards evaluating the robustness of neural networks. In 2017 ieee symposium on security and privacy (sp) (pp. 39-57). Ieee.

[2] Croce, F., & Hein, M. (2020, November). Reliable evaluation of adversarial robustness with an ensemble of diverse parameter-free attacks. In International conference on machine learning (pp. 2206-2216). PMLR.

**Final Justification:**

I have read all the comments from the other reviewers as well as the authors’ response. Most reviewers pointed out that the paper lacks a clear motivation and that the improvement over existing methods is marginal.

From my perspective, although the authors claim that their main purpose differs from that of the CW attack, the two objectives appear quite similar. In practice, attackers typically tailor their objective functions to the specific scenario. Therefore, I do not find the proposed objective function to be particularly novel. Likewise, I am not convinced that the halving rule used for adaptive step adjustment represents a new or original technique.

**Justification:**

I recommend a strong rejection.

This submission lacks clear novelty, and the motivation behind the proposed approach is not well articulated. Furthermore, based on the experimental results provided, there is insufficient evidence to demonstrate that the proposed methods are superior to existing alternatives.

---

> ### Author Rebuttal · Authors · 2025-10-22
>
> 1. We respectfully disagree with the assessment that our methods are minor variations of existing attacks. Our goal is not to compete with generic misclassification attacks but to extend adversarial robustness research into the underconfidence domain, a vulnerability class orthogonal to traditional accuracy focused threats. Equation 1 is not a reformulation of CW but a targeted loss for confidence suppression without misclassification. CW and similar optimization methods maximize cross entropy or margin losses that explicitly change predicted labels. In contract, our objective reduces the gap between the top-1 and top-2 probabilities while preserving the original class label, a property essential for evaluating calibration robustness as emphasized in previous work. ConfSmooth (Eq. 7 our attack) generalizes this idea by encouraging a near-uniform softmax distribution across all classes via label smoothing. This ensures broader uncertainty propagation, addressing the key limitation of existing calibration attacks that only manipulate two logits. Empirically, as shown in Table 1, ConfSmooth achieves stronger confidence suppression than prior dropout-based uncertainty attack across 14/15 dataset-architecture pairs, including ImageNet and medical imaging. These results demonstrate clear performance gains and conceptual novelty.
>
> 2. We agree that adaptive step-size mechanism exist in the literature, but our formulation serves a different purpose and domain. AutoAttack dynamically adjust step sizes to improve attack reliability for misclassification, whereas our step-back halving rule (Eqs. 5-6) ensures that the perturbed sample remains within the same decision region avoiding label flipper, a requirement unique to underconfidence attacks. This adaptive step-size variant thus prevents oscillations observed in the prior dropout-based attack (see Supplementary Fig. S1), where the predicted class alternates near decision boundaries. As Table 2 shows, the design increases perturbation efficiency while preserving label consistency, a property that existing adaptive attacks were never designed to enforce.
>
> 3. We appreciate the opportunity to clarify this point. The dropout mechanism is not applied during standard inference but within the perturbation generation process of the underconfidence attack, following Obadinma et al. (2024). Specifically, the dropout mask (Eq. 2) introduces stochastic sparsity into perturbations, effectively serving as a randomized search heuristic that reduces overfitting to gradient directions. This produces smoother perturbation distributions but also introduces instability (oscillations).
> Our ablation in Supplementary Section S1 shows that dropout-based attacks oscillate between class boundaries, while our adaptive variant removes this instability without needing dropout regularization. Thus, the dropout’s role is not standard test-time regularization but rather stochastic optimization within the attack generation process.
>
> 4. We agree that including additional strong baselines can improve completeness. Our evaluation focused on uncertainty-aware methods that were available at the time we were conducting the study. However, other methods such as AutoAttack as mentioned we can include it in the future to improve the completeness of this study.

---

### Official Review · Reviewer_ZkJV · 2025-10-09
**The paper needs to improve the experiments to support the effectiveness of the proposed attack and defense.**

**Rating:** 2
**Confidence:** 4

**Summary:**

This paper proposes a novel attack method for the underconfidence attack and derives a new defense by equipping the adversarial training with the proposed attack. First, the paper points out a problem with the existing underconfidence attack method. Specifically, the existing approach cannot prevent the repeated crossing of the decision boundary during iteration, resulting in oscillation between the two dominant output classes without considering other classes. To overcome this limitation, the paper proposes ConfSmooth, a new attack method that avoids boundary crossing by using an adaptive step size and encourages a uniform distribution to converge. Applying this attack method in the adversarial training setting, the paper also proposes a new defense method called Underconfidence Adversarial Training (UAT). Through experiments, the paper demonstrates the effectiveness of the proposed attack and defense and provides an additional assessment of the generated perturbations.

**Strengths:**

1. Compared to the misclassification attack, the underconfidence attack is a less-explored field, so we should encourage more research on the underconfidence attack.
2. The experiments used many datasets and baseline defense methods.

**Weaknesses:**

1. The motivation for encouraging near-uniform output distribution is unclear, especially if it significantly increases the visibility of perturbations. Assuming the model training has converged, it is unlikely that a model will output a near-uniform distribution given an input near the data manifold. Therefore, finding an example that induces a near-uniform output distribution could be even more challenging than finding a confidently incorrect adversarial example.
2. Also, the adaptive step size significantly increased the visibility of the generated perturbations. The proposed method would introduce overhead due to the step back, but Table 1 only shows marginal improvements. ConfSmooth shows better performance, but this is a natural consequence of pursuing a uniform distribution output and does not support the benefit of an adaptive step size. Even the dropout approach with ConfSmooth loss could achieve a better improvement than the adaptive attack.
3. The proposed method for adaptive step size may introduce overhead from the step back. This would increase the time required to generate the perturbation. This is another disadvantage and requires experiments to demonstrate the efficiency of the proposed method.
4. The proposed algorithm completely ruled out the dropout, but dropout could be a helpful component for a better output. At least, the dropout process helps reduce the perturbation size by retaining some elements, so the paper should have considered combining this into their algorithm after observing a large perturbation size.
5. With the increased perturbation magnitude (nearly 4 times the Dropout perturbation magnitude), the attack performance against adversarial training should not be considered an improvement.
6. The paper’s observation that “misclassification defense methods are vulnerable against underconfidence attacks” is not a non-trivial observation. While the misclassification defense only needs to preserve the correct label, the underconfidence defense requires maintaining high confidence in the correct label. Obviously, the latter is a harder requirement. For the same reason, “the underconfidence defenses achieving the misclassification robustness” is not an interesting finding, either. Defending with a stricter requirement (i.e., keeping high confidence in the correct label) can easily satisfy the less stringent requirement (i.e., preserving the correct prediction).

**Justification:**

To begin with, the motivation of this paper is not convincing enough. The authors point out the oscillations in the output probabilities (with two dominant prediction classes) during the iteration, but do not explain why such oscillations should be avoided. The experimental result actually demonstrated that the effort to prevent such oscillations significantly increases the visibility of the perturbations, which sounds like a shortcoming rather than a tradeoff. The improvement in attack performance is not impressive enough compared to the increase in perturbation magnitude introduced by the proposed method. The authors are confused about the difficulties between the misclassification defense and the underconfidence defense. As a result, they claim that the robustness of the proposed method to misclassification is interesting. Still, it should be interpreted as a consequence of the hardness of the underconfidence defense. Considering all of the above, I don’t find a meaningful value in this paper.

---

> ### Author Rebuttal · Authors · 2025-10-22
>
> 1. We appreciate the reviewer’s question regarding the motivation behind encouraging near-uniform output distributions. Our objective is not to force models into globally uniform outputs, but to simulate confidence erosion across all classes, a more realistic manifestation of underconfidence adversarial behavior. Existing underconfidence attacks manipulate only the top-2 logits, creating ambiguity between a pair of classes. However, this limited formulation fails to capture real-world calibration threats where uncertainty can spread broadly across classes (e.g., in multi-disease diagnostic systems). ConfSmooth addresses this by introducing controlled label smoothing (ε = 0.98, Eq. 7), which shifts the model’s predicted probability just enough toward a flatter distribution while preserving the top-1 label. This effect is local to the decision region, not globally uniform. As shown in Table 1 (p. 5) and Supplementary Fig. S2, ConfSmooth reduces confidence effectively without frequent misclassification, achieving its goal of uncertainty suppression.
>
> 2. We acknowledge that the adaptive step size increases the perturbation magnitude compared to Dropout, but this behavior is an intended trade-off to ensure label preservation. The dropout-based attack, while producing smaller perturbations, frequently crosses decision boundaries, leading to misclassifications and oscillations (as shown in Supplementary Fig. S1). The adaptive rule in Eqs. (5–6) explicitly mitigates this by reverting the step and halving α when a label flip occurs. This “step-back” operation preserves semantic correctness at the cost of slightly larger perturbations, a desirable outcome for underconfidence attacks that must not change the predicted label. We agree that ConfSmooth can achieve strong performance without an adaptive step, but the adaptive mechanism plays a crucial stabilizing role during training (UAT-Smooth). Indeed, Supplementary Table S3 demonstrates that UAT-Smooth with adaptive steps converges to optimal robustness in just 3–5 steps, achieving a balance between clean accuracy and defense robustness. This indicates the adaptive mechanism’s efficiency when integrated into the training framework.
>
> 3. We appreciate this concern and clarify that this step-back does not add significant overhead. The attack simply does not save the step that induces a misclassification and instead halves the step size.
>
> 4. We agree that dropout can be helpful as a stochastic regularizer for the attack, we can include this as a discussion point and run additional experiments to find the benefits.
>
> 5. We acknowledge that ConfSmooth produces stronger perturbations, as reported in Table 2, but this magnitude is a byproduct of distributing uncertainty over all classes, not an artifact of over-optimization. Importantly, ConfSmooth perturbations remain perceptually subtle (SSIM = 0.6136). Moreover, our robustness results (Table 3) demonstrate that even with these stronger perturbations, UAT-Smooth trained models maintain up to 0.57 confidence post-attack, indicating that the defense genuinely improves robustness, not merely because of stronger attacks. This supports that our method enhances defense generalization rather than exploiting perturbation strength.
>
> 6. We respectfully disagree that demonstrating robustness transfer between misclassification and underconfidence regimes is trivial.
> While it is intuitive that a defense maintaining confidence should also preserve accuracy, empirical results show that most calibration-oriented defenses (e.g., CAAT) fail to generalize this way. In contrast, UAT-Smooth does, achieving comparable PGD robustness to AT while halving gradient steps (Table 4). Thus, our finding that uncertainty-aware adversarial training confers robustness to both confidence and classification perturbations is not a tautology but a demonstrated cross-domain generalization, a property absent from prior work.

---

### Official Review · Reviewer_vvc8 · 2025-10-09
**Novel adversarial attack with benefits to robust training**

**Rating:** 5
**Confidence:** 4
**Final Rating:** 5
**Final Confidence:** 4

**Summary:**

The authors analyze shortcomings of prior uncertainty adversarial attacks and propose a more effective one (ConfSmooth) which maximizes the entropy of predicted label distributions.  Based upon ConfSmooth, the authors proceed to proposed an adversarial training scheme (UAT), which they show to improve robustness to both uncertainty attacks as well as misclassification attacks.

**Strengths:**

- A novel and effective adversarial attack combining cross-entropy and label smoothing to spread the uncertainty between target classes.
- An adversarial training scheme based on this new attack, demonstrating added robustness also to misclassification attacks.

**Weaknesses:**

- It's a bit confusing that the abstract mentions two proposed algorithm, with the first one being unnamed, and most of the attention going to the second one.  Indeed, the introduction only mentions ConfSmooth.  Overall, I can't tell the benefit of presenting two algorithms, with the discussion frequently alternating between the two.  I may be missing the punchline for the adaptive variant, but I encourage the authors to consider deferring most experiments on the adaptive variant to the appendix.

**Final Justification:**

Tightening up the presentation will help make the message more clear and effective.  I still encourage the authors to concisely summarize the insights from the first algorithm, deferring details to the appendix.

**Justification:**

The adversarial attack is well-motivated, clearly improves on prior methods, and its incorporation in adversarial training is shown to help develop more robust models.

---

> ### Author Rebuttal · Authors · 2025-10-22
>
> We sincerely thank the reviewer for the positive assessment of our work and for highlighting the contributions of ConfSmooth and UAT. We appreciate the constructive suggestion regarding the presentation of the two proposed attacks. Our paper introduces two complementary underconfidence attacks (an adaptive variant and ConfSmooth) serving distinct purposes:
>
> The Adaptive attack (Eqs. 4–6) focuses on optimization stability, addressing the oscillation issue in dropout-based calibration attacks (as shown in Supplementary Fig. S1). This variant is important conceptually because it provides the foundation for our defense algorithm (UAT) by ensuring that perturbations remain within the same decision region while lowering confidence.
>
> ConfSmooth (Eq. 7), built upon the adaptive variant, introduces label smoothing to spread uncertainty across all classes, yielding the strongest confidence suppression (Table 1) and forming the most effective attack for benchmarking.
>
> We agree that emphasizing both in the main text may cause confusion. In the revision, we will focus the narrative on ConfSmooth as the primary attack in both the Abstract and Introduction, since it is the conceptual and empirical backbone of the defense framework.

---

### Meta-Review · Area_Chair_XWQH · 2025-11-02

**Recommendation:** Accept (Poster)
**Confidence:** 4

**Metareview:**

## Summary

This work expands on existing adversarial attack paradigms by focusing on underconfidence attacks. The work presents two underconfidence attacks that introduces class ambiguity, and smoothens uncertainty across classes. Building on top of these attacks they formalize adversarial training. Experiments on multiple datasets and architectures shows improved performance under underconfidence attacks.

## Reviewer comments and rebuttal

The initial reviews received for this work were mixed, where two reviewers were critical of the motivation of the contribution. One of the reviewers characterizes the work to be reimplementations or minor variations of existing techniques. The authors clarify this point, and I am convinced by the authors; this work is discussing underconfidence attacks and is different from misclassification attacks that are more commonly studied in adversarial robustness literature.

Another reviewer rightly points out the large drops in clean accuracy. Authors attribute this to model capacity; I am not entirely convinced if this is the reason. I would encourage the authors to revisit this issue.

Overall, I think the focus on underconfidence attacks is convincing in this work. The two underconfidence attacks and the adversarial training show that maximum softmax probabilities (MSP) based underconfidence attack performance shows the methods in this work are sufficiently interesting for discussions.

---

### Decision · Program_Chairs · 2025-11-05

**Decision:**

Accept (Poster)

**Comment:**

We recommend a poster presentation given the AC and reviewers recommendations.